# Better Representations, Better Speech BCIs: a Multitask Approach

**Tommaso Boccato**
Tether Evo
tommaso.boccato@tether.io

**Michał Olak**
Tether Evo

**Jamie Oliver**
Tether Evo

**Chadwick Boulay**
Blackrock Neurotech

**Spencer Kellis**
Blackrock Neurotech

**Matteo Ferrante**
Tether Evo

## Abstract

Speech brain-computer interfaces represent a critical frontier for restoring communication in individuals with locked-in syndromes. While current approaches focus on scaling datasets and post-hoc language modeling, these strategies face immediate constraints due to data scarcity in neural recording settings. We propose a complementary path: enhancing representation learning through multitask training aligned with cortical encoding structure. Through systematic analysis of the largest intracortical speech dataset, we identify that acoustic features are most reliably encoded in speech motor cortex. Leveraging these insights, we develop multitask strategies combining phoneme prediction with acoustic regression and semantic objectives, achieving state-of-the-art performance: 13.7% word error rate. We also introduce the first end-to-end neural-to-sentence decoder for quasi-online communication. These results demonstrate that representation-first strategies substantially improve neural decoding within current data constraints while remaining compatible with future scaling efforts.

## 1 Introduction

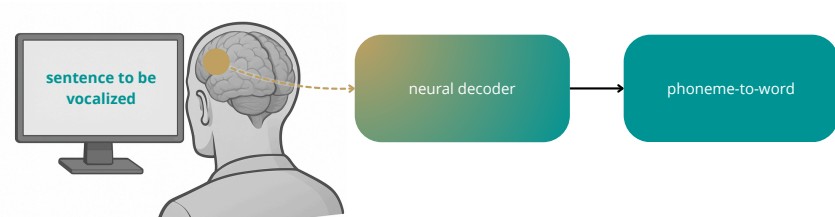

Figure 1: Overview of current speech decoding pipelines. A neural decoder maps signals from speech-related brain areas to an intermediate phoneme representation, which is subsequently translated into sentences.

Language serves as the primary medium of human communication, and its loss profoundly impacts quality of life through communication barriers and social isolation. For individuals who retain neural speech representations despite muscular impairments, neuroprosthetic systems offer the potential to restore communication by decoding neural signals into text or audio. Recent advances demonstrate feasibility using both non-invasive recordings (fMRI, MEG) [3, 17] and invasive approaches like multi-unit activity (MUA) [14]. While non-invasive methods show promise for semantic content

Preprint.

decoding, practical real-time communication requires word error rates (WERs) below 10% with minimal latency—performance currently achievable only through invasive recordings combined with deep learning architectures adapted from automatic speech recognition (ASR). At its core, neural speech decoding translates continuous neural signals into discrete word sequences, paralleling ASR development where breakthroughs emerged through dataset scaling and sophisticated architectures. However, neural decoding faces unique constraints: limited datasets due to recording complexity, electrode stability challenges, and signal nonstationarity over time.

**Evolution of neural speech decoding**    Breakthrough demonstrations have established the field's foundation. [12] first showed feasibility of real-time decoding with patients suffering from speech impairments using neural networks with n-gram language models. [20] achieved the first clinically viable performance by applying Connectionist Temporal Classification (CTC) to phoneme-level decoding, crossing into practical communication territory with error rates below 10% [14] on controlled vocabularies. The field then expanded toward naturalistic communication with [2] demonstrating direct speech synthesis from electrocorticography (ECoG) across diverse patients, and [9] introducing simultaneous text, audio, and visual avatar control while preserving voice characteristics. Latest systems [18, 8] achieve continuous large-vocabulary synthesis with conversational timing, reaching the goal of uninterrupted brain-to-voice communication under realistic interaction constraints. A table presenting a selection of recent studies on neural speech decoding can be found in Appendix A.

**Challenges and parallels with ASR development**    Despite remarkable progress, neural speech decoding faces fundamental constraints distinguishing it from traditional speech processing. The field remains fragmented due to experimental complexity requiring invasive recordings, multidisciplinary expertise, and long-term patient commitment. Most studies focus on single participants with laboratory-specific protocols, creating heterogeneity that complicates direct performance comparisons. More critically, current neural datasets remain orders of magnitude smaller than those driving ASR breakthroughs. While ASR benefits from over 500,000 hours of labeled data and sophisticated end-to-end architectures, neural decoding operates under severe constraints due to recording complexity and signal nonstationarity. This shapes the current technological landscape: successful systems consistently employ CTC-based phoneme decoders with external language models [21, 19, 1] (Figure 1) rather than the transformers defining modern speech recognition.

**Research questions and contributions**    Given current data limitations preventing direct application of modern end-to-end approaches, we argue that progress requires combining domain knowledge with principled, data-constrained strategies. Using the largest publicly available dataset [21], we address three key research questions: **(1)** What speech representations—articulatory, acoustic, or linguistic— are most reliably encoded in neural activity? **(2)** How can this knowledge inform architecture and training design? **(3)** Does multitask learning improve performance under data constraints? Previous work has shown that more data and better post-processing are key for successful brain decoding—both approaches falling under the philosophy of "more compute." Our main contribution traces a third path: better representation learning through multitask training. We demonstrate that **(1)** multitask training with acoustic and semantic constraints improves phoneme error rates, and **(2)** propose the first end-to-end model that directly decodes sentences in a quasi-online, direct setting, providing a framework for principled model development as the field matures toward larger-scale applications.

## 2    Methods

**Dataset**    We use the dataset from [20], comprising 9,000 neural recording trials from a participant with ALS across 24 recording days. Four 64-channel Utah microelectrode arrays (256 channels total) were implanted targeting ventral premotor cortex (area 6v) and Broca's area (area 44), with placement guided by individual fMRI activation and structural parcellation. The participant performed an instructed-delay task, attempting to speak sentences following visual prompts. Neural signals were preprocessed using threshold crossings in the ultra-high gamma band (250-5000 Hz) with a fixed threshold of -4.5 RMS, and spike band power was computed from the squared signal. Both features were binned in 20ms windows at 2kHz sampling rate. Day-specific normalization was applied to handle signal nonstationarities across sessions. We evaluate on 880 held-out test sentences with temporal separation from training data, ensuring realistic assessment under neural signal variability across time.

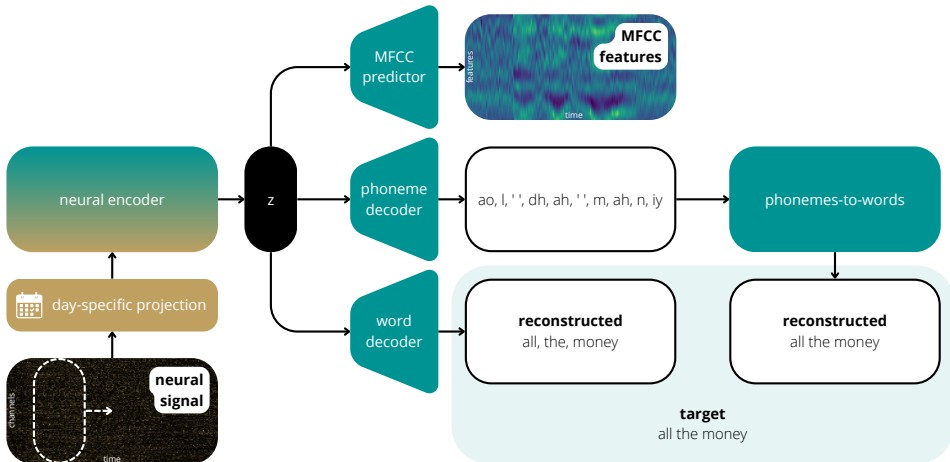

Figure 2: Schematic representation of our multitask decoder, consisting of a neural encoder that computes time-resolved neural embeddings ($z$), and three output heads predicting acoustic features, phonemes, and words.

**Neural encoding analysis**   To identify what speech information is encoded in neural activity, we performed encoding analysis using acoustic (mel-frequency cepstral coefficients - MFCCs), speech (Whisper embeddings), and semantic (BERT embeddings) features. Ridge regression models predicted features from neural data, revealing which representations are most reliably encoded across cortical regions (details in Appendix B).

**Architecture and training**   In the current data-limited regime, we prioritize representation learning over architectural complexity. Training strategy impacts performance more significantly than model expressivity when data is scarce. While ensembling and improved language modeling provide gains through computational scaling, we focus on enhancing learned representations by aligning objectives with neural encoding structure.

Our base architecture extends the five-layer bidirectional GRU from [20], which includes day-specific input transformations to handle neural nonstationarities and Gaussian noise injection for regularization. The model uses CTC loss to predict phonemes (40 classes: 39 ARPAbet phonemes plus silence) at 80ms intervals. Guided by the encoding model insights, we implement multitask training with auxiliary objectives aligned to neural representational content (Figure 2). We explore two auxiliary tasks: MFCC regression, where an additional linear head predicts 14 acoustic features to encourage alignment with time-resolved audio representations; and next-word prediction, integrating a pretrained BART decoder [7] with cross-attention over neural embeddings to incorporate sentence-level semantic information. In our experiments, we compare model variants with and without multitask objectives to evaluate whether auxiliary supervision improves phoneme sequence coherence and downstream word-level performance. Training details are provided in Appendix C.

**Phoneme-to-text conversion**   We compare two phoneme-to-text conversion approaches. The classical method uses weighted finite-state transducers (WFSTs) [10] combining phoneme posteriors with pronunciation lexicons and 5-gram language models, followed by optional neural language model rescoring. Alternatively, we explore end-to-end neural decoding that directly maps neural representations to sentences using pretrained language models, offering advantages in speed, efficiency, and long-range context modeling for real-time applications.

**Evaluation**   Performance is assessed using phoneme error rate (PER) and word error rate (WER) computed via edit distance on the held-out test set. This evaluation protocol enables direct comparison between WFST-based and end-to-end decoding strategies while maintaining realistic assessment conditions through temporal separation of training and test data.

# 3    Results

**Neural encoding analysis**    The encoding model analysis revealed distinct representational profiles across cortical regions (Figure 3 in Appendix D). Speech motor cortex showed strongest correlations with MFCC features, consistently outperforming synthetic speech features due to superior temporal alignment. Meaningful correlations with semantic and speech features emerged during preparatory phases, suggesting motor planning involvement. Broca's area exhibited lower but significant encoding performance across all feature types, indicating specialization for higher-level processing rather than direct articulation planning.

**Multitask training**    Multitask objectives systematically improved decoding performance (Table 1). Our GRU + CTC baseline achieved 18.1% PER, improving to 17.6% with MFCC regression and 16.9% with combined MFCC + next-word prediction—consistent with encoding findings showing MFCCs' strong neural correlation. Word-level performance followed similar trends: the multitask model with WFST decoding achieved 13.7% WER versus 17.4% baseline (Willett et al. [20]), while end-to-end BART decoding reached 24.2% WER (comparable to 23.8% online baseline from Willett et al. [20]). We additionally tested an RNN-T variant (Appendix E) of the proposed architecture that, however, consistently underperformed CTC-based models—likely due to the added model complexity outweighing potential benefits at the current data scale. A detailed analysis of how decoding performance scales with dataset size, instead, is provided in Appendix F.

Table 1: PER and WER comparison across baseline and proposed models.

| Model | PER (%) | WER (%) |
|---|---|---|
| Willett et al. (2023) [20] (125k words, improved LM) | 19.7 | 17.4 |
| Ours (GRU + CTC) | 18.1 | – |
|    + MFCC | 17.6 | – |
|    + MFCC + BART (offline, WFST and rescoring) | **16.9** | **13.7** |
| Willett et al. (2023) [20] (online) | – | 23.8 |
| End-to-end neural decoder (online) | – | 24.2 |

# 4    Discussion & conclusions

This study addresses a fundamental challenge in neural speech decoding: building practical neuroprostheses when data remains orders of magnitude smaller than in ASR. Our encoding analysis revealed that ventral premotor cortex reliably encodes low-level acoustic information (MFCCs), while higher-level representations remain weaker and more localized. Translating this insight into multitask training—combining CTC phoneme prediction with MFCC regression and next-word objectives—yielded state-of-the-art performance.

While Sutton's "Bitter Lesson" [16] emphasizes that scaling computation and data ultimately dominates handcrafted knowledge, current neural decoding operates in a unique interim regime where "representation-first" strategies can leverage cortical structure to boost data efficiency. Our comparative analysis reveals neural decoding following ASR's historical trajectory (Appendix G), but shifted three orders of magnitude toward the data-poor regime. RNN-Ts underperformed CTC baselines due to increased complexity exceeding current data benefits—mirroring early ASR development where simpler architectures preceded sophisticated models until sufficient data became available.

End-to-end neural language model decoding achieved near-instantaneous latency but incurred 10-15% WER penalty compared to WFST approaches, representing a critical speed-accuracy trade-off for real-time applications. Future research should focus on intent-based systems leveraging large language models, cross-subject foundation models for rapid calibration, and continual learning for electrode drift adaptation. Real-time BCIs also raise unprecedented privacy issues: decoding unspoken words presents risks for involuntary information leakage and unauthorized surveillance in deployed systems. Technical safeguards together with local-only inference are essential advances needed to prevent privacy violations. Finally, results from single-participant data may not generalize broadly, and multitask gains plateaued beyond 50% data availability. Nevertheless, we provide both a diagnostic

of current capabilities and a concrete roadmap for scaling neural speech decoding as larger datasets become available.

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

## A  Summary of recent neural speech decoding studies

Table 2 summarizes recent neural speech decoding studies involving invasive recordings. For each study, we report the task investigated (e.g., attempted speech, imagined handwriting), data preprocessing steps, the decoding target unit (e.g., phonemes, words), the model architecture used, dataset details including the associated vocabulary, and the reported performance.

## B  Neural encoding analysis

To investigate the information encoded in neural activity, we performed an encoding model analysis using three distinct types of target representations: low-level acoustic, high-level speech, and semantic features. Low-level acoustic features were computed as mel frequency cepstral coefficients (MFCCs) extracted from the participant's unintelligible speech audio. Even in this case, the dynamic of MFCC features could provide some information between word timings that could be used to better model neural dynamics. High-level speech features were derived from synthetic audio generated using the Suno Bark model [15] conditioned on ground truth transcripts; from this synthetic speech, we extracted contextual embeddings using the final hidden state of the Whisper encoder [13]. Semantic features were obtained as the pooled output of a pretrained BERT model [4] applied to the transcript text. Each neural trial was temporally aligned to the go-cue and segmented using a fixed window from $-1.0$s to $+2.0$s (total duration of 3.0s). The neural data, originally sampled at 2kHz, were smoothed and downsampled using average pooling to obtain a final resolution of 200 time bins per trial. We then trained ridge regression models to predict each dimension of the target features (MFCC, Whisper, or BERT) from the corresponding neural data using 10-fold cross-validation. Encoding performance was quantified by computing the pearson correlation across samples for each channel and time-point.

To assess statistical significance, we estimated a null distribution by randomly permuting the correspondence between neural activity and target representations across trials, repeating the encoding procedure over 1000 permutations. This procedure controls for spurious correlations and allows us to determine whether the observed correlation values significantly exceed chance levels. Significant encoding was defined as model performance exceeding the 99[th] percentile of the null distribution. This analysis provides insights into which neural features and cortical locations encode low- or high-level speech and language representations during attempted speech production.

## C  Training details

We trained all models for 100 epochs with a batch size of 64, using the Adam optimizer [6] and a learning rate of 1e-4. The loss term weights were assigned as follows: 1 and 1 for the CTC + MFCC setup, 1 and 0.1 for the CTC + BART setup, and 0.1, 0 and 1 for the CTC + MFCC + BART setup. Models trained using the BART objective (CTC + BART and CTC + MFCC + BART) were fine-tuned from those obtained through CTC and CTC + MFCC, respectively. Training was conducted on a machine equipped with 512GB of RAM and NVIDIA H100 GPUs.

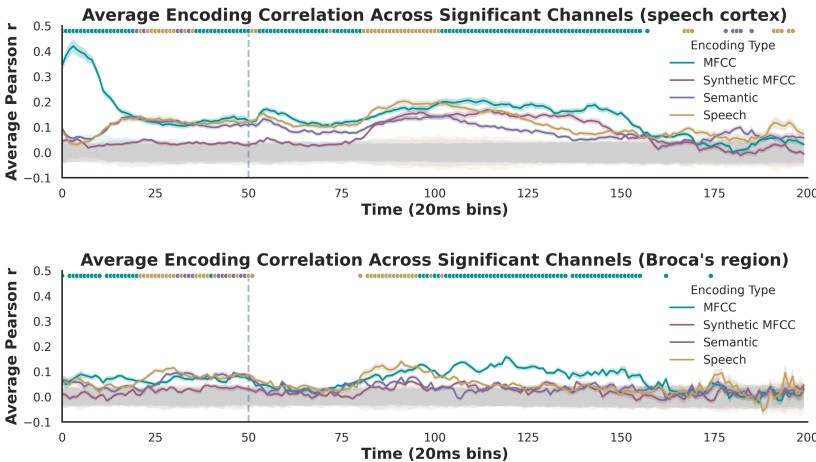

Figure 3: Encoding model performance, in terms of Pearson correlation between predicted and true features, across cortical regions and feature types. **Top**: speech cortex. **Bottom**: Broca's region. Each curve shows the mean ±SEM over time (20ms bins). Colored dots above the plot indicate significant time steps, defined as those exceeding the 99th percentile of a null distribution obtained from 1,000 encoding models trained on permuted neural and language features.

## D    Neural encoding analysis results

In this section, we present a visualization (Figure 3) of the results obtained from the neural encoding analysis.

## E    RNN-Transducers

To address CTC's independence assumption, we investigated RNN-Transducer (RNN-T) architectures [5], which jointly model acoustic and linguistic information. The RNN-T framework uses three components: a transcription network (bidirectional GRU in our case), a prediction network (4-layer RNN functioning as phoneme-level language model), and a joiner network (2-layer feedforward) that combines outputs for context-dependent phoneme predictions. This approach potentially reduces dependence on external language models by incorporating temporal dependencies directly in the neural decoder. However, we observed that this model consistently underperforms compared to the GRU variant across all configurations, yielding a PER of around 20% regardless of the use of auxiliary losses (20.4% for the baseline RNN-T model, 20.3% with the MFCC auxiliary loss, and 20.2% with the MFCC + BART variant). We attribute this behavior to the limited scale of the available data, which prevents effective training of the phoneme-level prediction network—a key component of the RNN-T framework—thereby reducing the potential benefits of auxiliary losses.

## F    Impact of training set size on model performance

To evaluate how model performance scales with the size of the training set, we conducted a systematic scaling analysis by training models on increasing fractions of the available dataset. Specifically, we trained models using randomly sampled subsets comprising 1%, 5%, 10%, 25%, 50%, 75%, and 100% of the total training trials, corresponding to durations ranging from a few minutes to approximately 14h of neural recordings. Results are reported in Figure 4.

Across all model variants and training conditions, performance exhibited a logarithmic growth pattern, with diminishing returns as more data were added. Importantly, we observed a consistent crossover point around 50% of the training data, beyond which multitask models began to reliably outperform their single-task counterparts. This effect was evident across both PER and WER metrics, suggesting that auxiliary objectives become more effective once a sufficient quantity of training data is available.

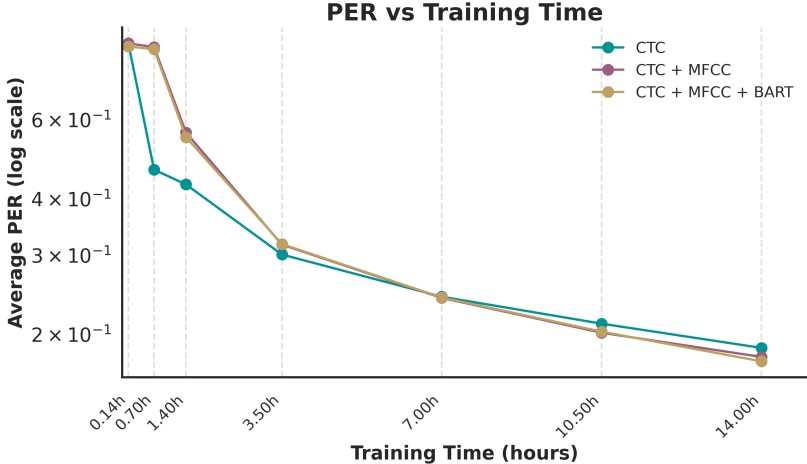

Figure 4: Scaling behavior of decoding performance w.r.t. dataset size. Performance improves logarithmically as the dataset grows. Multitask training begins to consistently outperform the single-task baseline when more than 50% of the available data is used.

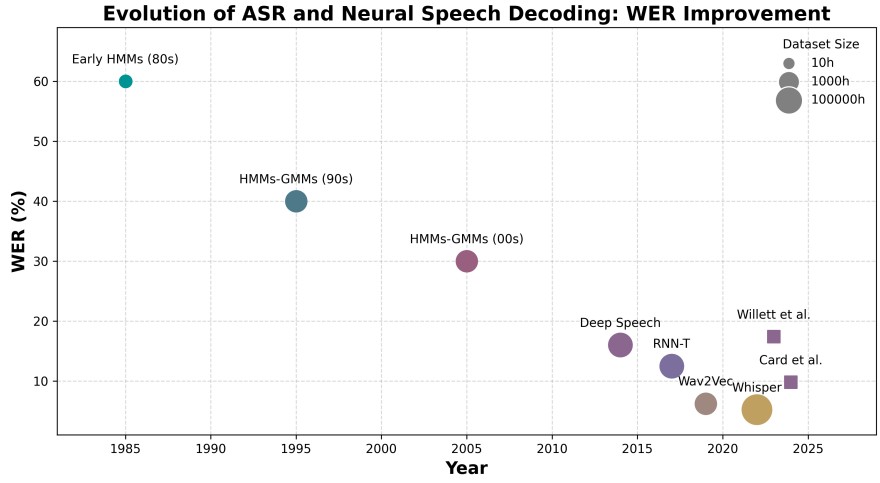

Figure 5: Progress in ASR and neural speech decoding over the past four decades. Circle sizes reflect the dataset sizes used for training (log scale). CTC-based models are represented in lilac. Years and WERs should be considered approximate.

These findings indicate that multitask learning—specifically, augmenting phoneme classification with auxiliary objectives such as MFCC regression or next-word prediction—yields small but reliable gains in decoding accuracy in data regimes that support more complex representational learning. We hypothesize that as dataset sizes continue to grow, the benefits of multitask supervision may become increasingly pronounced, offering regularization and inductive biases that promote the learning of more robust and generalizable neural representations.

# G ASR and neural speech decoding historical trajectories

Figure 5 illustrates how advancements in ASR and neural speech decoding have influenced performance in each field, by showing their respective progress over time in terms of WER.

Table 2: Recent neural speech decoding studies.

| Reference | Recording type | Brain area(s) | Task | Preprocessing | Target | Model | Dataset size | Vocabulary | Performance |
|---|---|---|---|---|---|---|---|---|---|
| Moses et al. (2019) [11] | ECoG | Motor speech areas | Q&A | Spike-rate features | Answer from set of candidates | NN + n-gram LM | 1 patient, 48 sessions (22h) | 50 words | 25% WER, 15.6 WPM |
| Moses et al. (2021) [12] | Intracortical | Motor speech areas | Attempted speech | Spike-rate features | Words | RNN + n-gram LM | 1 patient, 48 sessions (22h) | 50 words | 25% WER, 15.6 WPM |
| Willett et al. (2021) [19] | Intracortical | Premotor cortex | Im. handwriting | Spike activity | Characters | RNN + LM | 1 patient, cont. writing | Alphabet + symbols | 94.1% acc., 90 CPM |
| Willett et al. (2023) [20] | Intracortical | Brodmann 6v, 44 | Attempted speech | Spike counts and band power | Phonemes, words | RNN + CTC + n-gram LM | 1 patient, 10k sentences | 50 words / open vocab. | 9.1% WER (50 words), 23.8% WER (open), 60+ WPM |
| Metzger et al. (2023) [9] | ECoG (253 channels) | Sensorimotor and temporal speech areas | Attempted speech | Spike features | Text, audio, avatar | RNN + CTC + n-gram LM + vocoder | 1 patient, multiple sessions | Open vocab. | 25% WER, 78 WPM, real-time avatar |
| Card et al. (2024) [1] | Intracortical (256 channels) | Left ventral precentral gyrus | Attempted speech | Spike counts and band power | Phonemes, words | NN + vocoder | 1 patient, 84 sessions (2h/16h) | 50 words / 125,000 words | 99.6% acc. (50 words), 90.2% acc. (125k words), 97.5% acc. (stable), 32 WPM |
| Chen et al. (2024) [2] | ECoG | Bilateral | Speech | Spike features (70-150Hz) | Speech waveform | CNN or transformer + synthesizer | 48 patients, multiple sessions | 50 words | Pearson $r$ ~0.80 |
| Wairagkar et al.(2025) [18] | Intracortical (256 channels) | Ventral precentral cortex | Attempted speech | Spikes counts and band power (600ms windows) | 20D acoustic vec. | Transformer + vocoder | 1 patient, 4.1k-8.3k trials, 956 sentences | Open vocab | 96.4% acc., 0.83 spectrogram corr., 10ms latency |
| Littlejohn et al. (2025) [8] | ECoG (253 channels) | Speech cortex | Silent speech | Spike counts and band power (80ms windows) | Text, audio stream | RNN-T + n-gram LM + vocoder | 1 patient, multiple sessions | 50 phrases / 1,024 words | 47.5-78 WPM, ~1s latency |

