# OpenReview forum: "Better Representations, Better Speech BCIs: a Multitask Approach"
_NeurIPS.cc/2025/Workshop/UniReps — UniReps2025_

### Official Review · Reviewer_EY5C · 2025-09-11
**Neat technique, incomplete comparisons and SoTA claims, unclear fit to workshop**

**Confidence:** 5

**Review:**

**Quality:** The submission makes principled changes to an existing speech decoding model, i.e., adding additional training objectives, in order to improve overall decoding performance. The authors have justified their modifications through an analysis of the neural data. The method leads to performance improvements over the popular, performant GRU baseline – however, comparisons to other baselines from the Brain2Text '24 benchmark are missing and claims of state-of-the-art performance are incorrect.

**Clarity:** The submission is well-written for the most part and has neat figures and results, however, some details are sparse (see Suggestions section).

**Originality:** The approach is original in that predicting MFCCs and words in addition to phonemes has not been explored on this dataset, to my knowledge. Otherwise, it has been observed in prior neural decoding work that multi-task decoding can improve performance [1].

**Significance:** At the current scale of data for neural speech decoding, such approaches could be very impactful in revealing the importance of certain inductive biases and/or regularisation strategies to improve performance. However, given other more performant related work which has not been compared against, it is unclear how significant this work will be,

**Relevance:** I'm quite unsure about the relevance of this submission to the UniReps workshop. While the correlation between MFCCs and attempted speech-related motor activity is interesting, adding an objective to predict MFCCs from a model's latents does not necessarily make the model's representations similar to the brain's (in fact, neural activity inputs to the model are in some ways the brain's representations themselves). Furthermore, it seems like the MFCC and word prediction losses could just be serving as some form of regularisation to prevent overfitting here given the (relatively) small size of the dataset and known overfitting issues with the GRU baseline. So, overall, I think it is unclear how well this paper fits in the overall theme of representation alignment and other main topics of interest (in fact, I believe there were other workshops where this would be far more relevant).

**Strengths:**
* The paper is mostly well-written and the experiments seem principled given the advances in ASR and evidence of multitask pre-training improving neural decoding [1].
* The performance improvements from the method over the GRU baseline are quite good, especially given the simplicity of adding objectives.

**Weaknesses & Suggestions:**
* Glaringly, the claims of state-of-the-art performance are untrue and overstated, both in terms of phoneme error rate (PER) and word error rate (WER). As shown in the Brain2Text '24 benchmark paper [2], DCoND-LIFT [3] obtained a PER of ~15% (lower than this submission's 16.9%) while their WER was ~5.77% (versus 13.7% here). Subsequently, recent work [4] has further improved PER to ~14% with a causal, near-real-time Transformer. Thus, I believe the authors should reevaluate their claims here.
* It is unclear to me what the authors mean by next word prediction. The dataset, to my knowledge, only has sentence-level labels and no dense word-wise annotations. How do the authors decide what the "next" word is for a given segment? What is the exact objective being used?
* Unlike other recent work [4,5], this method is not well-suited to online/real-time inference due to the use of a non-causal, bidirectional GRU backbone.
* Furthermore, the method is not easily scalable to additional data from, e.g., more days due to training a day projection layer that needs to be relearnt for each day. The authors could consider removing this layer to improve parameter-efficiency and have better scalability to additional data: in my experience with the current dataset and the GRU baseline, the day projection layer isn't particularly necessary to achieve good performance.
* Finally, the claims of biological motivation (e.g., in the discussion) seem tenuous at best to me – training to predict MFCCs because they are well-correlated with some neural activity does not strike me as biological inspiration for the architecture design or choice of objectives.

**Assessment:** Overall, while the work is a neat idea and demonstration, I don't think the paper meets the bar for inclusion due to the lack of proper comparisons with prior work [2], slightly improper claims, and unclear relevance to the workshop. However, this is an important line of work and I encourage the authors to continue exploring whether their additional objectives improve existing, performant methods even further.

**References:**
1. Azabou, Mehdi, et al. "Multi-session, multi-task neural decoding from distinct cell-types and brain regions." The Thirteenth International Conference on Learning Representations.
2. Willett, Francis R., et al. "Brain-to-Text Benchmark'24: Lessons Learned." arXiv preprint arXiv:2412.17227 (2024).
3. Li, Jingyuan, et al. "Brain-to-text decoding with context-aware neural representations and large language models." Journal of Neural Engineering (2024).
4. Feghhi, Ebrahim, et al. "Time-Masked Transformers with Lightweight Test-Time Adaptation for Neural Speech Decoding." arXiv preprint arXiv:2507.02800 (2025).
5. Hee-Woon Ryoo, Avery, et al. "Generalizable, real-time neural decoding with hybrid state-space models." arXiv e-prints (2025): arXiv-2506.

**Score:**

1

**Topic Fit:**

1

---

### Official Review · Reviewer_qVZN · 2025-09-15
**Review of Better Representations, Better Speech BCIs: a Multitask Approach**

**Confidence:** 3

**Review:**

The paper promotes representation first learning rather than scaling data and language models to address data scarcity in invasive speech BCIs. Using an intracortical dataset from an ALS participant over 24 days, encoding analyses show that the speech motor cortex captures acoustics MFCCs, while Broca’s area carries weaker planning and semantic signals. The authors train a multitask decoder with a bidirectional GRU and CTC plus MFCC regression and next word prediction. Multitask training reduces PER and WER, reaching 13.7 percent WER with WFST. A direct neural-to-sentence path yields 24.2 percent WER.

The main strengths of this paper include:
(1) The paper introduces a promising encoding analysis of the auxiliary tasks (MFCC and next-word), aligning supervision with what cortex actually encodes rather than relying on architectural tweaks alone.
(2) The paper demonstrates strong absolute gains (WER down to 13.7%) in an invasive, single-participant, long-horizon setting; the end-to-end decoder is a meaningful step toward low-latency BCI use.
(3) The paper presents a day-specific normalization, strict temporal train/test splits, multi-head decoding, and comparisons against WFST and neural baselines, making the evidence persuasive.

We have the following major comments:
(1) We are concerned that the findings may not generalize across participants, implants, or tasks; cross-subject transfer or few-shot calibration would strengthen claims about generality.
(2) It is nice to formulate the next-word objective leverages BART. However, there is little analysis disentangling structure captured in neural embeddings from what is offloaded to the language model (e.g., representational similarity or probing of the shared subspace).
(3) The authors are suggested to include confidence intervals or paired bootstrap tests for headline PER/WER ; the significance of the 13.7% versus 17.4% gap should be quantified, ideally with effect sizes.
(4) The authors claim that state-of-the-art references a particular benchmark configuration; an apples-to-apples scoreboard using the same vocabulary, language model, and held-out days would better situate the gains.
(5) The result of quasi-online 24.2% WER is promising, but latency profiling (token-emission delay, jitter) and failure-mode analysis are needed to assess real-time viability.

**Score:**

3

**Topic Fit:**

3

---

### Official Review · Reviewer_DbGh · 2025-09-15
**A modest step-forward in using representation-first approaches under data constraints**

**Confidence:** 3

**Review:**

**Summary**

This paper investigates strategies for improving neural speech decoding under severe data limitations, focusing on representation learning through multitask training. Using intracortical recordings from a single participant with ALS, the authors perform encoding analyses to identify which acoustic and semantic representations are most reliably encoded in cortical regions. They then propose a multitask decoder that integrates phoneme prediction (via CTC), acoustic regression (MFCCs), and semantic context (next-word prediction with a pretrained BART decoder). Results show modest but consistent improvements in phoneme and word error rates (WER) over a GRU+CTC baseline, with state-of-the-art WER of 13.7% under a WFST-based pipeline, and a first demonstration of quasi-online end-to-end decoding with a WER of 24.2%. The authors argue that these findings suggest a “representation-first” pathway forward in an era where larger datasets are not yet feasible.

**Quality and Clarity**

The paper is overall well written and clearly motivated: the authors explain the central challenge (data scarcity in neural speech decoding) and why representation-focused approaches may be more fruitful than simply scaling models. This serves as a good baseline for their work and why it is important. Dataset description (24 days, ~9,000 trials, temporal split) and preprocessing steps are detailed, and the multitask architecture is broken down into CTC phoneme decoding, MFCC regression, and next-word prediction. There are some clarity issues with the timing inconsistencies when describing the neural data (20 ms bins, 80 ms steps, 200 bins over 3s), but the methods otherwise seem clear. Also, towards the end, there is a claim of "near-instantaneous latency" but this is a qualitative description, and numbers would help clarify this. Since this workshop focuses on representations, interpretability, and their role in advancing intelligence, this paper seems to be a good fit for interesting discussions relating to the role of representation learning in better neural decoding methods.

**Originality**

The novelty seems to lie in two main contributions: Neuroscience-guided auxiliary tasks, with MFCCs and semantic embeddings chosen based on encoding analyses of motor cortex activity, and a demonstration of an end-to-end neural-to-sentence decoder with access to limited data. Instead of proposing any radically new concepts, the work seems to combine multiple different methods from within the field to address a specific issue.

**Significance**

While modest when looking at it in scale, the results seem promising, with the WER of 13.7% using the WFST pipeline certainly being an improvement upon prior results (17.4%). Practically, it improves data efficiency and pushes the benchmark forward. However, the scope is limited by a single participant, absence of statistical significance tests, and lack of analysis of model failures. These constraints temper the broader significance but still make the work a solid step forward.

**Pros**

- Clear framing of the challenge (data scarcity).
- Thoughtful connection between neuroscience findings and auxiliary task design.
- Concrete preliminary results on a benchmark dataset.
- Introduction of quasi-online decoding as a proof-of-concept.
- Strong fit for representation-focused workshop discussions.

**Cons**

- Limited statistical rigor (no confidence intervals, modest improvement sizes).
- Scope limited to one participant and one dataset.
- Novelty of auxiliary tasks could be better distinguished from prior multitask approaches.

**Score:**

4

**Topic Fit:**

2

---

### Official Review · Reviewer_4Ax6 · 2025-09-16
**Review of submission 35**

**Confidence:** 3

**Review:**

Summary: This paper analyses speech data to identify which features are most significant and reliably encoded. The insights from this analysis are used in a multitask training process for the proposed method. The novel method is evaluated and compared against the state of the art.

Strengths:
- The authors provided a thorough and easy-to-read review of prior work and their limitations.
- The dataset, model architecture, and experimental setup are clearly described in a reproducible manner.
- This work addresses an important open challenge in representation learning for BCI.

Weaknesses:
- The experiment results (Table 1) should include confidence intervals or similar to convey statistical significance.
- It would be nice to see results or figures associated with the neural encoding analysis. The insights gained from this analysis are key to the proposed method. While details about the analysis are provided in words in the appendix, I feel it would be better to be in the main text, as it is a key result/insight.

I suggest including the following citation as well:
"Neuro-MoBRE: Exploring Multi-subject Multi-task Intracranial Decoding via Explicit Heterogeneity Resolving", Wu et al.

Overall, this paper is very well written and the work is a significant contribution to the field.

**Score:**

4

**Topic Fit:**

2